# Effect of Root-Knot Nematode Disease on Bacterial Community Structure and Diversity in Peanut Fields

**Lijun Wu [1], Yan Ren [1], Xiangsong Zhang [2], Guanghui Chen [1], Chuantang Wang [1], Qi Wu [1], Shuangling Li [1], Fudong Zhan [1], Li Sheng [3], Wenliang Wei [4,*] and Mei Yuan [1,*]**

[1] Key Laboratory of Peanut Biology, Genitics and Breeding/Ministry of Agriculture and Rural Affairs, Shandong Peanut Research Institute/Shandong Academy of Agricultural Sciences, Qingdao 266100, China; 15863444111@163.com (Y.R.)
[2] Linyi Agricultural Technology Extension Center, Linyi 276001, China
[3] Qingdao Academy of Agricultural Sciences, Qingdao 266100, China
[4] College of Agriculture, Yangtze University, Jingzhou 434000, China
[*] Correspondence: whwenliang@163.com (W.W.); yuanbeauty@126.com (M.Y.)

**Abstract:** The root-knot nematode (RKN) disease is a highly destructive soilborne disease that significantly affects peanut yield in Northern China. The composition of the soil microbiome plays a crucial role in plant disease resistance, particularly for soilborne diseases like RKN. However, the relationship between the occurrence of RKN disease and the structure and diversity of bacterial communities in peanut fields remains unclear. To investigate bacterial diversity and the community structure of peanut fields with severe RKN disease, we applied 16S full-length amplicon sequencing based on the third high-throughput sequencing technology. The results indicated no significant differences in soil bacterial α-diversity between resistant and susceptible plants at the same site. However, the Simpson index of resistant plants was higher at the site of peanut-wheat-maize rotation (Ro) than that at the site of peanut continuous cropping (Mo), showing an increase of 21.92%. The dominant phyla identified in the peanut bulk soil included *Proteobacteria*, *Acidobacteria*, *Actinobacteria*, *Planctomycetes*, *Chloroflexi*, *Firmicutes*, and *Bacteroidetes*. Further analysis using LEfSe (Linear discriminant analysis effect size) revealed that *Sulfuricellaceae* at the family level was a biomarker in the bulk soil of susceptible peanut compared to resistant peanut. Additionally, *Singulisphaera* at the genus level was significantly more enriched in the bulk soil of resistant peanut than that of susceptible peanut. Soil properties were found to contribute to the abundance of bacterial operational taxonomic units (OTUs). Available phosphorus (AP), available nitrogen (AN), organic matter (OM), and pH made a positive contribution to the bacterial OTUs, while available potassium (AK) made a negative contribution. The metabolic pathway of novobiocin biosynthesis was only enriched in soil samples from resistant peanut plants. Eleven candidate beneficial bacteria and ten candidate harmful strains were identified in resistant and susceptible peanut, respectively. The identification of these beneficial bacteria provides a resource for potential biocontrol agents that can help improve peanut resistance to RKN disease. Overall, the study demonstrated that severe RKN disease could reduce the abundance and diversity of bacterial communities in peanut bulk soil. The identification of beneficial bacteria associated with resistant peanut offered the possibility for developing biocontrol strategies to enhance peanut resistance to RKN disease.

**Keywords:** peanut (*Arachis hypogaea* L.); root-knot nematode disease; bulk soil; bacterial community structure

## 1. Introduction

*Arachis hypogaea* L. (peanut or groundnut), commonly known as peanut or groundnut, holds great economic importance as both economic crop and oil crop across the world [1]. It is cultivated worldwide in tropical and subtropical regions. Taxonomically, peanut is considered a legume and is believed to have originated in Central and South America,

with cultivation spread to other parts of the world [2]. In many countries, peanut provides a significant nutritious contribution to the diet due to their rich protein, lipid, and fatty acid content [3], and peanut is grown for oil production, peanut butter, confectionary, snacks, and protein extenders globally [4]. Plant-parasitic nematodes are considered one of the major obstacles to the production of peanut crops [5], and one of the most influential nematodes is the root-knot nematode [6].

The root-knot nematode (*Meloidogyne hapla* Chitwood, RKN) can cause significant yield losses in the cultivated peanut and has become an important factor influencing peanut production in Northern China [7]. On the one hand, different peanut varieties have different resistance to RKN. Most peanut cultivars are highly susceptible to RKN [8]. It is well known that the wild diploid peanut relatives showed strong resistance to RKN [9]. But our preliminary tests found that HY9810 was more resistant to RKN than HY20 (Figure 1). They can be used as a good material to study peanut RKN disease. On the other hand, different planting methods have a great influence on RKN. Long-term continuous cropping of peanuts will undoubtedly aggravate the occurrence of RKN disease [10], and the traditional approach tends to use crop rotation to control RKN [11].

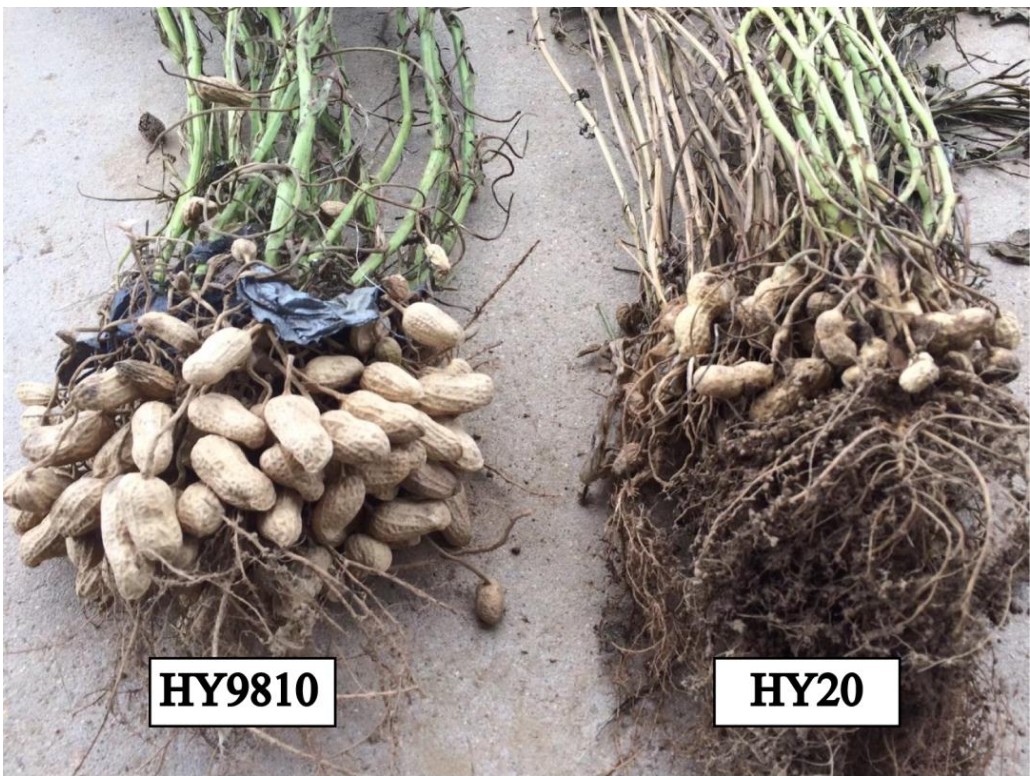

**Figure 1.** Photographs of HY9810 (R) and HY20 (S) at harvest time at the Mo. Site.

Bacterial community and structural changes are closely related to RKN disease. Bacteria are the most abundant and widely distributed microorganisms in soil. They play an important role in maintaining the healthy decomposition of organic matter in the soil, promoting material circulation, and maintaining the balance of the soil ecosystem [12]. Cao et al. observed that RKN infection changed the α-diversity and microbial composition of root microorganisms and drove the transformation of microorganisms [13]. Lu et al. found that the community structure and function of the plant rhizosphere were significantly correlated to the RKN disease [14]. Li et al. demonstrated that community variation and assembly of root endophytic microbiota were significantly affected by RKN [15]. Rani et al. revealed that the bacterial bioagents, namely *B. amyloquefaciens*, *B. megaterium*, *P. fluorescens*, and *P. putida*, showed the potential for controlling RKN [16].

In general, different peanut genotypes and cropping patterns had a great influence on RKN disease. The interaction of bacteria−pathogens is the theoretical basis for improving colonization and controlling the effect of biocontrol bacteria [13]. To explore the relationships between bacterial communities, soil environments, and plant health, bacterial communities were analyzed using the third high-throughput sequencing technology of the 16S full-length rDNA amplicons in peanut samples with different peanut genotypes and cropping patterns. It would provide a theoretical basis for the exploitation and utilization of microbial resources for controlling RKN disease in peanut.

## 2. Materials and Methods

### 2.1. Soil Samples Collection and Processing

Two peanut germplasms (HY9810 and HY20) and two planting sites were chosen in this experiment. HY9810 (R-) was an advanced line developed by the disease-resistant breeding team in Shandong Peanut Research Institute (SPRI), which is resistant to RKN (Figure 1). HY20 (S-) was released in 2002 by SPRI, which is susceptible to RKN (Figure 1). One of two planting sites called Mo (35°30′13″ N, 119°11′43″ E) was located at Jiajiagou, Shanzhuan Town, Rizhao City, Shandong Province, and the soil type was sandy loam; another one designated as Ro (36°48′37″ N, 120°30′03″ E) was an experimental base of SPRI, located at Wangcheng Town, Laixi City, Shandong Province, and the soil type was sandy loam. Peanut was planted continuously for seven years at the Mo site which had severe RKN disease. Peanut, wheat, and maize were planted alternatively at the Ro location which had light RKN disease. The two peanut varieties were sown in May 2019. In mid-September, the bulk soils of peanut fields were collected by the five-point sampling method. In each site, the surface soil was removed, and five soil cores at a depth of 5~20 cm near the peanut plants were collected and mixed into one bulk soil. Twelve composite samples were obtained by repeated sampling three times. The samples were sieved through two mm mesh and divided into two groups, with one group stored at room temperature for measurement of soil physicochemical properties, and the other group frozen at −80 °C for DNA extraction of bacteria community. Mo.R and Mo.S stand for soil samples of HY9810 and HY20 from the field of severe RKN disease, respectively. Ro.R and Ro.S were soil samples of HY9810 and HY20 from fields of light RKN disease, respectively.

### 2.2. 16S rDNA Full-Length Amplification and Sequencing

The soil bacterial diversity and community structure were detected by 16S rDNA full-length sequencing. All the operation processes, including total soil DNA extraction, amplification, library construction, and sequencing, were performed by Novogene (Beijing, China); data analysis was carried out by Gene Denovo (Guangzhou, China). Total genome DNA from soil samples was extracted by the CTAB method. DNA concentration and purity were determined by 1.0% agarose gel and ultraviolet spectrophotometry [17]. The DNA was diluted with sterile water to 1 ng/L, according to the concentration. The specific primers were 16S F (forward primer, 5′-CCTACGGGNGGCWGCAG-3′) and 16S R (reverse primer, 5′-GACTACNVGGGTATCTAATCC-3′) with barcode [18]. The amplified library was sequenced using a PacBio SMRT RS II DNA sequencing platform (Pacific Biosciences, Menlo Park, CA, USA). Low quality was filtered by PacBio circular consensus sequencing technology [19], and the chimera sequences were removed [20] using the UCHIME algorithm [21]. Sequences with ≥97% similarity were assigned to the same OTU (Operational taxonomic unit) by analyzing sequences performed by Uparse software (Uparse v7.0.1001, http://drive5.com/uparse/ accessed on 26 September 2021) [22,23]. The species that were selected to rank top 10 in terms of mean abundance in all samples were visualized using stacked plots, other known species were categorized as others, and unknown species were marked as unclassified with the R project ggplot2 package (version 2.2.1). To identify differences of bacterial communities among the four soil groups, Venn diagrams were plotted with the VennDiagram package [24]. The PCA analysis was performed on the community composition structure of four soil groups at the OTU level to reduce the dimen-

sion of the original variables to explore the similarity and differences among groups using the QIIME software package, v. 2 [25]. UPGMA (Unweighted pair-group Method with Arithmetic Mean) is a commonly used clustering analysis method, which mainly refers to the hierarchical clustering analysis method using any distance to evaluate the similarity among the soil groups [25,26]. A species accumulation boxplot was used to investigate the species composition of a sample and predict species abundance in a sample [27]. LEfSe (LDA Effect Size) analysis was used to find biomarkers with statistical differences between groups [28,29], and the Lefse analysis parameter set to the alpha value of factorial Kruskal Wallis test between classes was <0.05, and the threshold value of logarithmic LDA score for distinguishing features was >2.0. Multiple direct gradient regression was used to analyze the correlation between microflora and environmental factors based on soil basic physical and chemical properties and OTU annotation data. The R software language Vegan package, v. 2.6-4, was used for Canonical correlation analyses (CCA), and Pearson, the maximum correlation coefficient between environmental factors and differences in sample community distribution, was used to judge the significance of CCA analysis. Spearman rank correlation was used to study the relationship between environmental factors and bacterial species richness to obtain the correlation and significant $p$-value between each other. Based on the species and environmental factors, the R language vegan package was used for variance partitioning analysis (VPA) of the contribution (percentage) of each group of environmental factor variables to the species distribution. The relative abundance of screened beneficial and harmful bacteria is displayed using the R language circlize package.

### 2.3. Determination of Soil Physical and Chemical Properties

Soil basic physicochemical properties of each sample were determined, including alkali-hydrolyzed nitrogen (AN), available phosphorus (AP), available potassium (AK) [30], organic matter (OM) [31], and pH [32]. AN content was determined by the alkali diffusion method [33]. AP content was determined by the molybdenum antimony colorimetric method with a UV-visible spectrophotometer (Shimadzu UV-2700, Kyoto, Japan) [34]. The AK was digested by $CH_3COONH_4$ and measured by flame atomic absorption spectrophotometry with flame spectrometry (Sherwood M410 Britain, Sherwood scientific Ltd., Cambridge, UK) [33]. OM content was analyzed using dichromate oxidation [31]. Soil pH was measured with a pH meter at a soil to water ratio of 1:2.5 (Meter 3100C, Licor, Lincoln, NE, USA) [33].

### 2.4. Statistical Analysis

Analysis of variance (ANOVA) with Least-Significant Difference (LSD tests) and Tukey's HSD test were applied to distinguish significant differences between each treatment. All statistics were carried out in IBM-SPSS 22.0 software, and significance was set at $p < 0.05$.

## 3. Results

### 3.1. Bacterial Community Diversity of Bulk Soil in Peanuts Field

#### 3.1.1. Sequence Data of 16S Full-Length rDNA

To investigate the bacterial structure and diversity in peanut bulk, we sequenced the 16S full-length amplicon of 12 soil samples. Low-quality reads were corrected or removed, SSR filtered, primers removed with the Cutadapt software v. 4.4, and chimera sequences removed to obtain high-quality reads. Overall, a total of 164,184 raw reads were obtained; 162,706 high-quality clean reads were finally obtained (Table S1), which were used to cluster the analysis of operational taxonomic units (OTUs) with 97% identity (Figure 2).

A total of 3739 OTUs were found in high-quality reads, belonging to 2 domains, 25 phyla, 62 classes, 121 orders, 172 families, 360 genera, and 547 species (File S1). A total of 258 OTUs were common among the 4 groups of soil samples, while 220, 138, 287, and 249 OTUs were unique to Mo.R, Mo.S, Ro.R, and Ro.S, respectively. A total of 53 OTUs were uniquely found in Mo.R and Ro.R, and 61 OTUs were uniquely found in Mo.S and Ro.S (Figure S1).

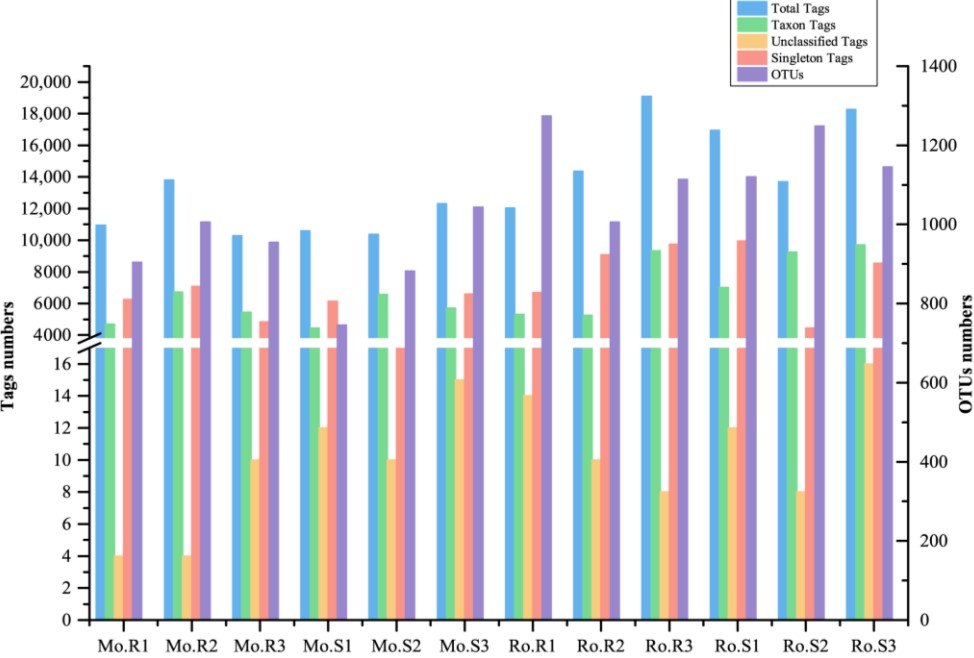

**Figure 2.** The column chart of preprocessing statistics and quality control of the data.

### 3.1.2. Alpha Diversity

Through alpha diversity analysis, the community diversity of bacteria was examined by the Simpson and ACE indices. The larger the Simpson implies higher species diversity [35]. The Simpson index analysis showed an extremely significant difference ($p < 0.01$) and ACE indices showed a significant difference ($p < 0.05$) between the Ro site and Mo site; no significant difference in the alpha diversity index was observed between different genotypes ($p > 0.05$) (Table S2). The box plot of α diversity index was drawn and the significance of the difference between every two groups was done by Tukey's HSD test (Figure 3). A comparative analysis of the four groups using the Simpson index revealed that the Mo.R increased by 1.06% compared to Mo.S, the Ro.R increased by 0.15% compared to Ro.S, the Ro.R increased by 1.13% compared to Mo.R, and the Ro.S increased by 2.05% compared to Mo.S. A comparative analysis of the ACE of the four groups revealed that the Mo.R increased by 0.36% compared to Mo.S, the Ro.R increased by −5.61% compared to Ro.S, the Ro.R increased by 21.92% compared to Mo.R, and the Ro.S increased by 29.64% compared to Mo.S (Figure 3 and Table S3).

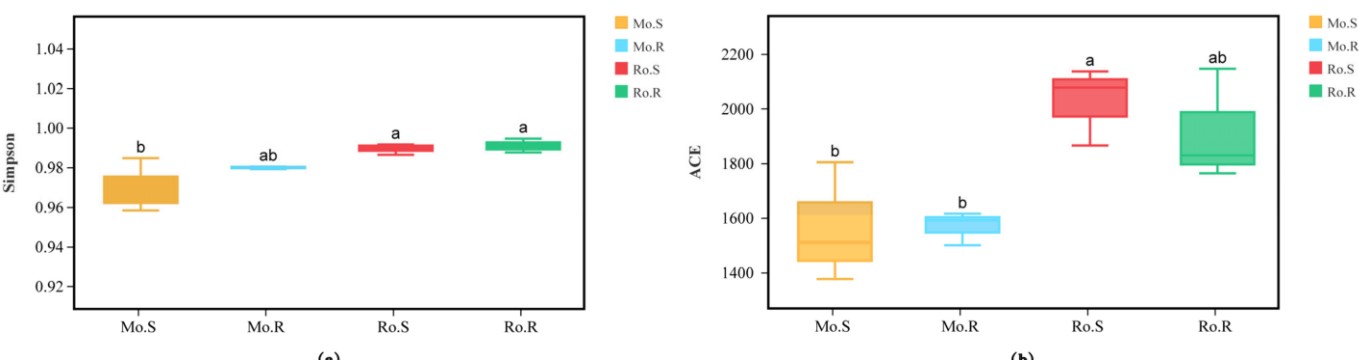

**Figure 3.** Box plot of alpha diversity in four groups. (**a**) Simpson index. (**b**) ACE Index. Different lowercase letters indicate significant difference at $p < 0.05$.

### 3.1.3. Beta Diversity

Principal component analysis (PCA) and clustering analysis were used to observe the similarities among the four soil groups. The first two principal components (PC1 and PC2) of PCA explained 59.02% and 18.22% of the total variation, respectively (Figure 4a). Cluster analysis also revealed that soil samples from the same planting sites were classified into a group (Figure 4b).

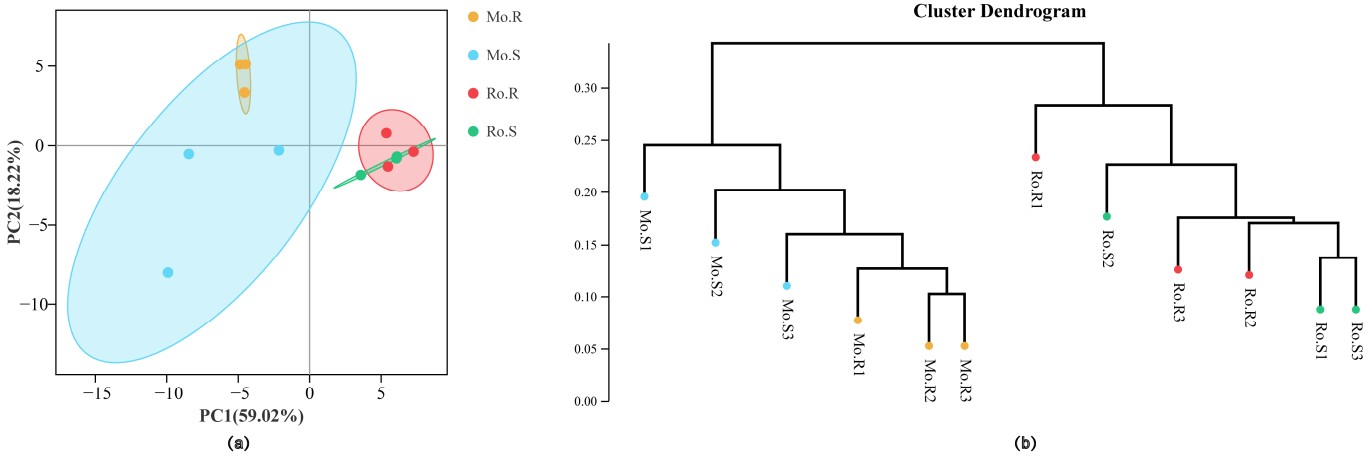

**Figure 4.** Beta diversity analysis. (**a**) Principal component analysis (PCA) analysis at the OTU level. The same color points belong to the same soil group. (**b**) UPGMA (Unweighted Pair-Group Method with Arithmetic Mean) at the family level.

### 3.1.4. Bacterial Community Structure of Peanut Bulk Soil Influenced by RKN

In order to further analyze the structure of the bacterial community, the abundance distribution of each group at the levels of phylum, class, order, family, genus, and species was shown according to the results of taxonomic annotation (Figure 5).

All OTUs were classified into 25 phyla, and there were 8 phyla in each group with a relative abundance >1% (Table S4). *Proteobacteria*, *Acidobacteria*, *Actinobacteria*, *Planctomycetes*, *Chloroflexi*, *Firmicutes*, and *Bacteroidetes* were the seven dominant phyla in the peanut bulk soil, accounting for about 90% of all bacterial taxa in each group (Figure 5a). Compared to Mo.S, the relative abundance of *Actinobacteria*, *Planctomycetes*, and *Chloroflexi* increased by 1.12%, 38.22%, and 3.24% in Mo.R, respectively; compared to Ro.S, that increased by 14.84%, 16.96%, and 20.39% in Ro.R, respectively. Conversely, compared to Mo.S, the relative abundance of *Proteobacteria*, *Firmicutes*, *Bacteroidetes*, and *Nitrospirae* decreased by 8.46%, 12.17%, 53.34 and 8.11% in Mo.R, respectively; compared to Ro.S, that decreased by 2.14%, 13.51%, 6.40% and 18.23% in Ro.R, respectively (Figure 5a).

At the class level, most of the bacteria belonged to *Acidobacteria*, *Gammaproteobacteria*, *Alphaproteobacteria*, *Ktedonobacteria*, *Rubrobacteria*, *Betaproteobacteria*, *Planctomycetia*, *Phycisphaerae*, *Acidimicrobiia*, and *Chitinophagia* (Table S5). The most dominant bacterial populations in Mo.R, Mo.S, Ro.R, and Ro.S accounted for 80.84%, 80.98%, 72.49%, and 76.34%, respectively. Compared to Mo.S, the relative abundance of *Ktedonobacteria*, *Planctomycetia*, and *Phycisphaerae* increased by 2.01%, 48.40%, and 15.46% in Mo.R, respectively; compared to Ro.S, that increased by 14.57%, 9.07% and 23.64% in Ro.R, respectively. In contrast, compared to Mo.S, the relative abundance of *Gammaproteobacteria*, *Alphaproteobacteria*, and *Chitinophagia* decreased by 21.33%, 4.67%, and 65.56% in Mo.R, respectively; compared to Ro.S, that decreased by 18.21%, 1.57% and 11.80% in Ro.R, respectively (Figure 5b).

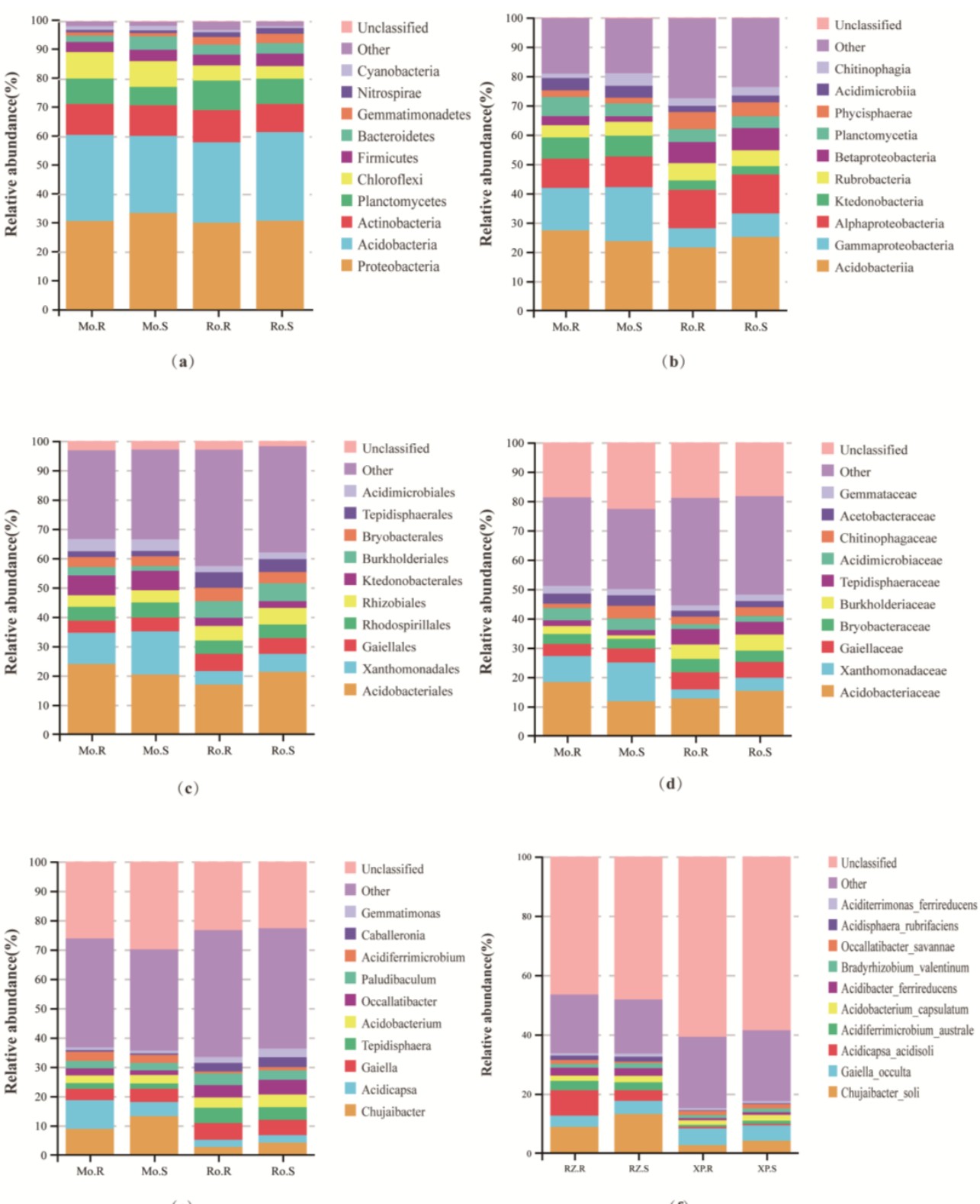

**Figure 5.** The top 10 species distribution in the average abundance of bacterial communities in peanut bulk soil at the levels of phylum (**a**), class (**b**), order (**c**), family (**d**), genus (**e**), and species (**f**). Other known species are classified as others, and unknown species are marked as unclassified.

Acidobacteriales was the most abundant bacterial order, which accounted for 23.96%, 20.34%, 16.97%, and 21.27% in Mo.R, Mo.S, Ro.R, and Ro.S, respectively (Figure 5c). Compared to Mo.S, the relative abundance of *Ktedonobacterales*, *Bryobacterales*, and *Tepidisphaerales*

increased by 1.27%, 1.67%, and 8.79% in Mo.R, respectively; compared to Ro.S, that increased by 20.76%, 18.19% and 21.76% in Ro.R, respectively. While compared to Mo.S, the relative abundance of *Xanthomonadale*, *Rhodospirillales*, and *Rhizobiales* decreased by 27.69%, 8.61%, and 4.41% in Mo.R, respectively; compared to Ro.S, that decreased by 25.41%, 1.84%, and 11.90% in Ro.R, respectively (Figure 5c).

*Acidobacteriaceae* and *Xanthomonadaceae* were the most abundant families in the four groups collectively (Figure 5d). Compared to Mo.S, the relative abundance of *Bryobacteraceae* and *Tepidisphaeraceae* increased by 1.67% and 8.79% in Mo.R, respectively; compared to Ro.S, that increased by 18.81% and 21.76% in Ro.R, respectively. In contrast, compared to Mo.S, the relative abundance of *Xanthomonadaceae*, *Chitinophagaceae*, and *Acetobacteraceae* decreased by 33.04%, 65.64%, and 4.27% in Mo.R, respectively; compared to Ro.S that decreased by 30.14%, 12.39%, and 2.86% in Ro.R, respectively (Figure 5d).

A thorough investigation at the genus level showed that 360 taxa were classified from the 4 bulk soil communities, whereas most genera were <15%, implying high bacterial diversity in the 4 soil groups (File S2). As shown in Figure 5e, the predominant identifiable genera were *Acidicapsa*, *Chujaibacter*, *Gaiella*, and *Occallatibacter*, which accounted for 9.80%, 13.21%, 5.69%, and 5.02% in Mo.R, Mo.S, Ro.R and Ro.S, respectively. Compared to Mo.S, the relative abundance of *Tepidisphaera* increased by 9.13% in Mo.R; compared to Ro.S, that increased by 20.83% in Ro.R. In contrast, compared to Mo.S, the relative abundance of *Chujaibacter*, *Acidobacterium*, and *Gemmatimonas* decreased by 32.99%, 9.85%, and 2.03% in Mo.R, respectively; compared to Ro.S, that decreased by 35.65%, 19.58% and 30.22% in Ro.R, respectively (Figure 5e and Supplementary File S2).

*Chujaibacter soli* was the most abundant species in Mo.R and Mo.S, which accounted for 8.85% and 13.21%, respectively; *Gaiella occulta* was the most abundant species in Ro.R and Ro.S, which accounted for 5.65% and 5.17%, respectively (Figure 5f). Compared to Mo.S, the relative abundance of *Acidicapsa acidisoli* increased by 137.21% in Mo.R; compared to Ro.S, that increased by 19.89% in Ro.R. While compared to Mo.S, the relative abundance of *Chujaibacter soli*, *Acidobacterium capsulatum*, *Bradyrhizobium valentinum*, and *Acidisphaera rubrifaciens* decreased by 32.96%, 18.17%, 25.67%, and 13.49% in Mo.R, respectively; compared to Ro.S, that decreased by 35.73%, 23.63%, 18.90%, and 4.86% in Ro.R, respectively (Figure 5f).

### 3.2. Analysis of the Microbiological Biomarkers in the Peanut Bulk Soil

In order to analyze the biomarkers between different groups, LEfSe (LDA Effect Size) analysis was employed in the four groups of peanut bulk soil. Statistical analysis was performed from the phylum to the genus level in cladograms, and LDA scores of 2 or greater were confirmed by LEfSe between resistant and susceptible peanut (Figures 6 and S2). As can be seen from Figure 6, 17 biomarkers were pointing to susceptible peanut, while *Chujaibacter* and *Xanthomonadaceae* had LDA scores ≥ 4.0, and 43 biomarkers pointing to resistant peanut, while *Planctomyceteria*, *Acidobacteria*, *Acidicapsa*, *Acidobacteriaceae*, and *Acidobacteriia* had LDA scores ≥ 4.0 at the Mo site; 17 and 20 biomarkers were pointing to susceptible and resistant peanut at the Ro site, respectively.

*Sulfuricellaceae* at the family level as the specific biomarker was both pointing to susceptible peanut at the same site, which suggested that *Sulfuricellaceae* can be used as biomarkers of the susceptible peanut. *Singulisphaera* at the genus level as the specific biomarkers were both pointing to resistant peanut at the same site, which suggested that *Singulisphaera* can be used as biomarkers of resistant peanut.

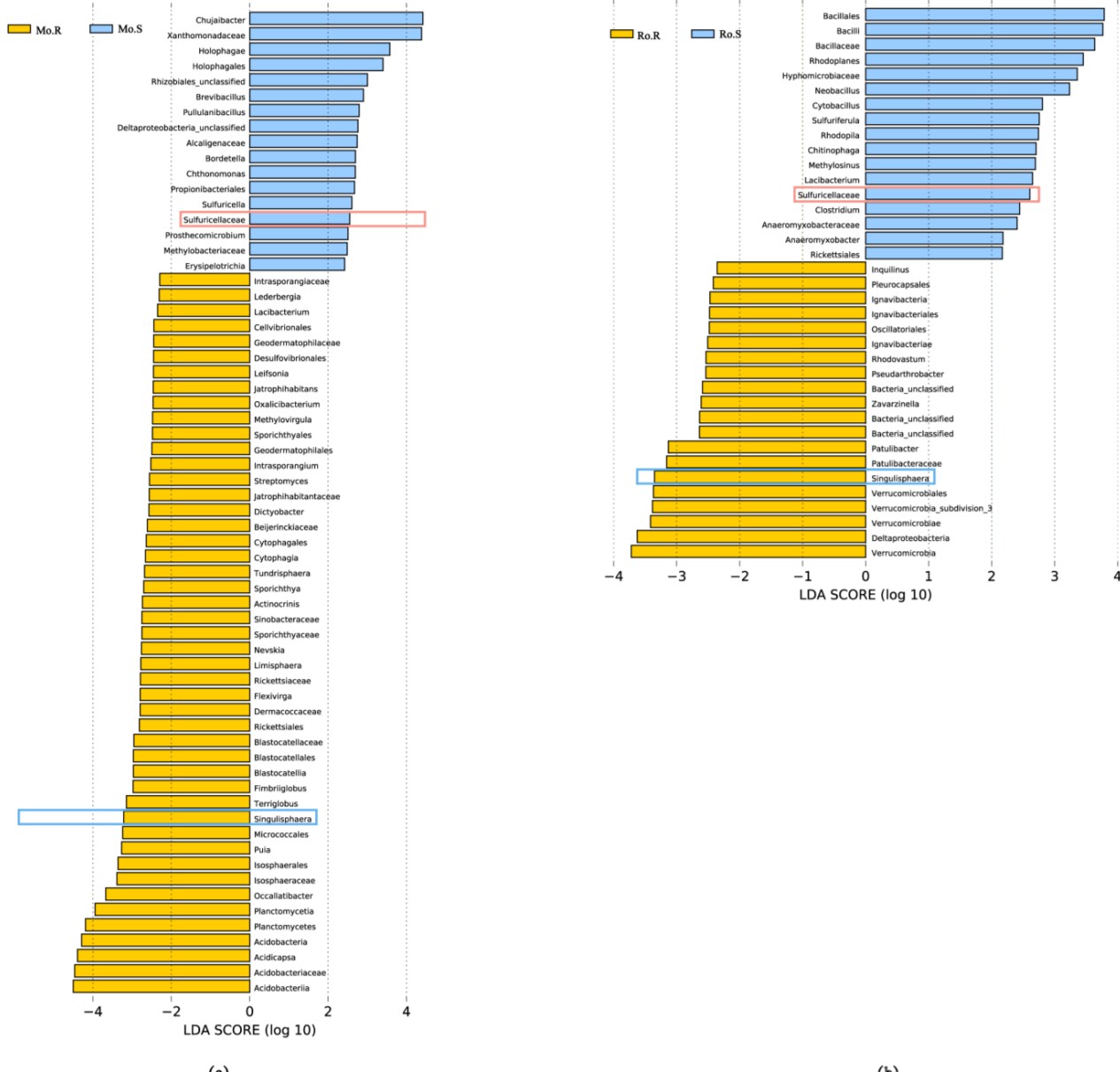

**Figure 6.** LEfSe Bar of different abundance between resistant and susceptible peanut at Mo site (**a**) and Ro site (**b**) by linear discriminant analysis (LDA). The yellow horizontal bars are the biomarkers enrichment in bulk soil of RKN-resistant peanut, which had a negative LDA score; the blue horizontal bars are that of RKN susceptible peanut, which had a positive LDA score. The blue frame box is the common biomarker of RKN resistant peanut. The red frame box is the common biomarker of RKN susceptible peanut.

### 3.3. Relationship between Bacterial Community Structure and Environment Factors in Peanut Bulk Soil

Spearman correlation analysis was used to study the relationship between the composition of bacterial community structure and environmental factors. Soil physical and chemical factors were determined, including pH, organic matter (OM), alkali-hydrolyzed nitrogen (AN), available phosphorus (AP), and available potassium (AK) (Table 1). The results showed that the levels of OM, AN, and AP at the Mo site was higher than that at the Ro site, while the pH of the Ro site was higher than that at the Mo site. However, no significant patterns were observed between the soil physicochemical traits and the susceptibility or resistance of the peanut.

**Table 1.** Environmental chemical characteristics in four groups of peanut bulk soil.

| Item | Mo.S | Mo.R | Ro.S | Ro.R |
|---|---|---|---|---|
| pH | $4.5 \pm 0.1$ a | $4.6 \pm 0.1$ a | $4.8 \pm 0.2$ a | $5.1 \pm 0.2$ a |
| Organic matter (OM, g/kg) | $13.5 \pm 0.7$ a | $14.9 \pm 0.7$ a | $12.1 \pm 0.3$ a | $10.9 \pm 0.7$ a |
| Available nitrogen (AN, mg/kg) | $82.9 \pm 10.8$ a | $85.5 \pm 10.4$ a | $60.7 \pm 5.7$ a | $54.6 \pm 6.3$ a |
| Rapidly available phosphorus (AP, mg/kg) | $103.4 \pm 8.2$ a | $121.1 \pm 13.3$ a | $94.3 \pm 1.3$ a | $81.1 \pm 6.3$ a |
| Available potassium (AK, mg/kg) | $46.1 \pm 0.0$ a | $53.6 \pm 14.2$ a | $61.2 \pm 3.3$ a | $42.3 \pm 6.5$ a |

Note: The same lowercase letters in the table indicate that the differences of each item among different groups are not significant $p > 0.05$.

Multiple direct gradient regression was used to analyze the relationships among sampling points, microflora, and environmental factors, and canonical correspondence analysis (CCA) was constructed. As shown in Figure 7a, the descending order of influences on the distribution of bacterial species in the peanut bulk soil were OM, AP, pH, AN, and AK (the corresponding r2 were 0.8326, 0.8002, 0.6000, 0.7220, and 0.1579, respectively). OM, AP, and AN had extremely significant effects (Pr < 0.01), pH had significant effects (Pr < 0.05), but AK had no significant effects (Pr > 0.05). AP, AN, OM, and pH made a positive contribution to the OTUs of bacterial, while AK made a negative contribution (Figure 7b).

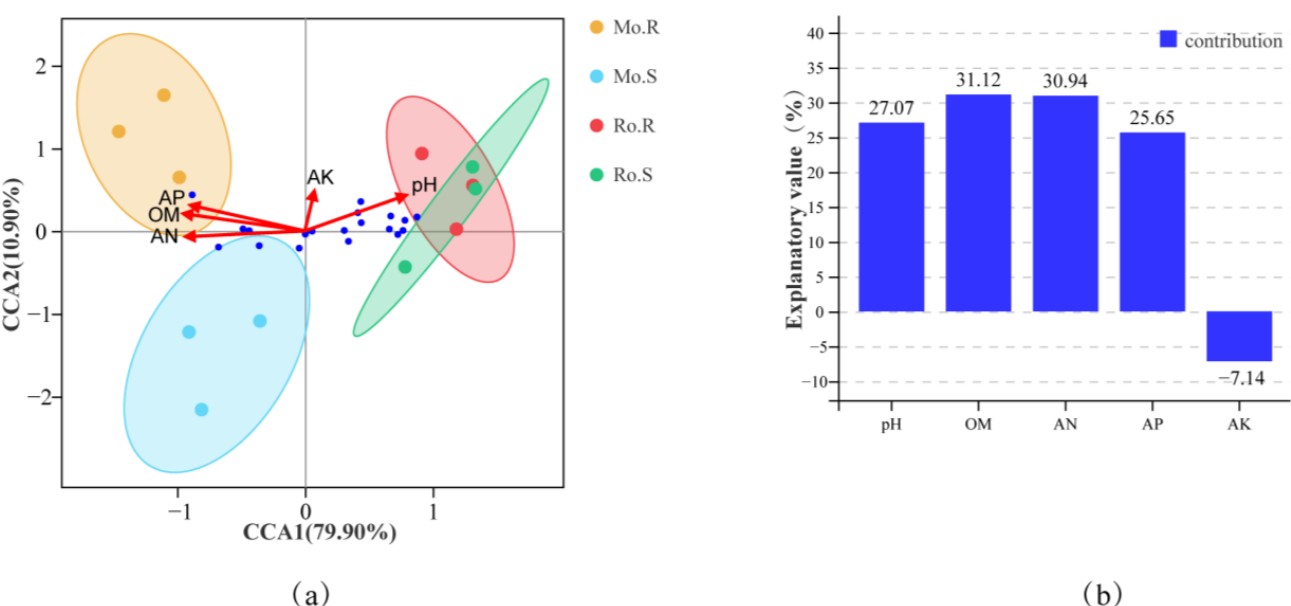

**Figure 7.** (**a**) Canonical correspondence analysis (CCA) of species information between the peanut bulk soil and environmental factors. Blue spots indicate the top 20 bacterial species. (**b**) Contribution map of environmental factors.

Spearman correlation analysis was performed to study the mutual change relationship between environmental factors and species based on the measured data of soil environment factors and the OTU data of each sample. As seen in Figure 8, OM was the most significant environmental factor in the top 20 of the bulk soil bacteria at specific levels; next were AN and AP, and the last was pH. Moreover, the four environmental factors showed an extremely significant correlation with the abundance of *Acidibacter ferrireducens*, *Acidicapsa acidisoli*, and *Acidiferrimicrobium australe*.

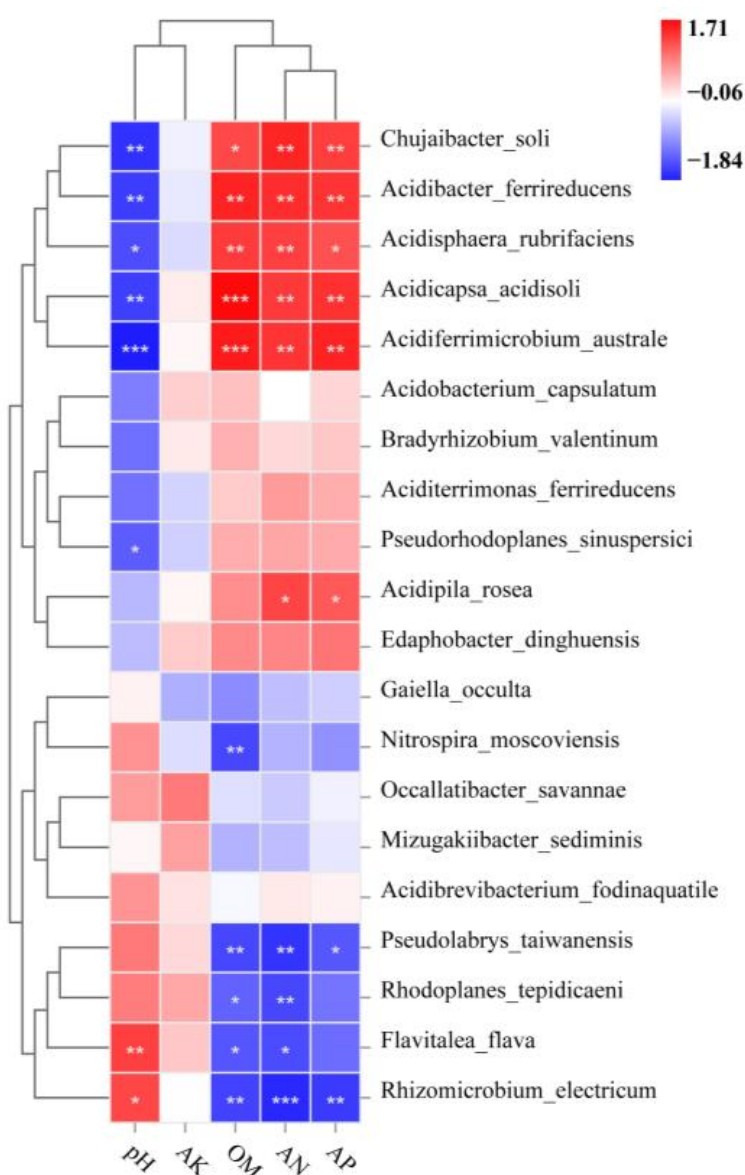

**Figure 8.** Heat map of Spearman correlation analysis between environmental factors and bacterial species. Colors indicate the strength of the correlation, * indicates a significant correlation with a *p*-value less than 0.05, ** indicates a highly significant correlation with a *p*-value less than 0.01; *** indicates *p*-values less than 0.001.

### 3.4. Analysis of Beneficial and Harmful Bacteria in the Peanut Bulk Soil

Beneficial bacteria are defined as those that can promote plant growth, help prevent pathogen invasion, and improve plant adaptability to abiotic or biological stresses; they are also called plant growth-promoting rhizobacteria (PGPR) [36,37]. In order to further analyze the bulk soil bacterial diversity of R-cultivars on the basis of relevant literature studies, 11 bacterial species were chosen to draw a Circos diagram. The results showed that the relative abundance of RKN-resistant peanut was higher than that of RKN susceptible peanut (Mo.R-Mo.S $\geq$ 0 and Ro.R-Ro.S $\geq$ 0) (Figure 9a). Among them, the large proportion included *Burkholderia cepacia*, *Jatrophihabitans soli*, *Arthrobacter dokdonellae*, *Rhodanobacter lindaniclasticus*, and *Nitrosospira multiformis*. The relative abundance of *Mucilaginibacter ximonensis* and *Ferruginibacter alkalilentus* in susceptible and resistant plants at the same planting site was a leap from absence to present, although the difference in value was not significant.

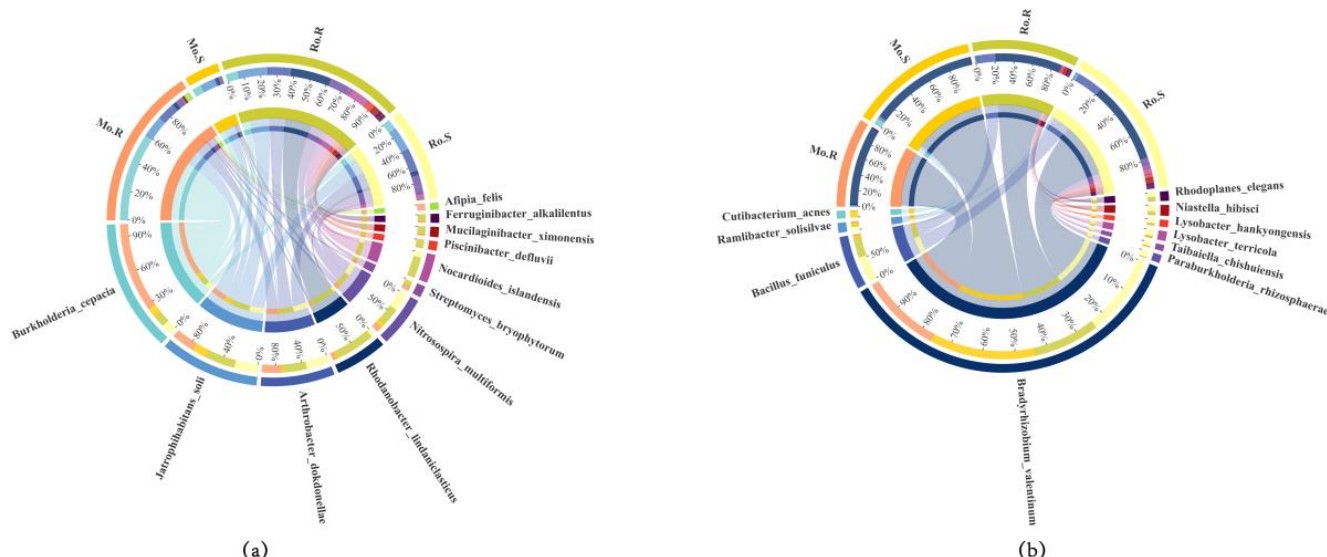

**Figure 9.** The Circos map of candidate beneficial and harmful bacteria resistant to RKN in peanut bulk soil; (**a**) beneficial bacteria, (**b**) harmful bacteria.

Harmful bacteria are defined as the opposite of beneficial bacteria, which inhibit plant growth, help pathogens to invade, and reduce plant adaptation to abiotic or biotic stresses [38,39]. To further analyze the bulk soil bacterial diversity of S-cultivars, 10 bacterial species were chosen to draw the Circos diagram, which indicated that the relative abundance of RKN-susceptible peanut was higher than that of RKN-resistant peanut (Mo.R-Mo.S ≤ 0 and Ro.R-Ro.S ≤ 0) (Figure 9b). The large proportions were *Bradyrhizobium valentinum*, *Bacillus funiculus*, *Niastella hibisci*, *Ramlibacter solisilvae,* and *Rhodoplanes elegans.* It is worthy of note that the relative abundance of *Cutibacterium acnes* from resistant to susceptible peanut in the same planting site was a qualitative leap from nothing to something, although there was little difference in value.

### 3.5. Prediction of Bacterial Functional Potential in the Peanut Bulk Soil

To explore the functional roles of bacteria in peanut bulk soil, PICRUSt2 was used to predict their function based on KEGG metabolic pathways and the relative frequencies of predicted functions. A total of 171 metabolic pathways in KEGG were annotated (File S3). Among the 6 primary pathways, the number of metabolic pathways annotated was at most (118); the next was genetic information processing annotated to 18; the third was cellular processes annotated to 9. In the second level of the metabolic pathway, Xenobiotics biodegradation and metabolism, Carbohydrate metabolism, and the metabolism of terpenoids and polyketides were the 3 most annotated pathways, with 17, 15, and 14 annotated pathways, respectively. Some representative pathways were selected for demonstration and analysis. Seven pathways were both increased in Mo.R compared to Mo.S and Ro.R compared to Ro.S (Figure 10). These seven metabolic pathways mainly focus on the first level of Genetic information processing, Human diseases and metabolism, and the secondary level of Transcription, Translation, Immune diseases, Biosynthesis of other secondary metabolites, and Metabolism of terpenoids and polyketides. Three of the seven metabolic pathways belong to the Metabolism of terpenoids and polyketides. Interestingly, the abundance of novobiocin biosynthesis was over 300 in both resistant peanut, but 0 in both susceptible peanut at the same planting site.

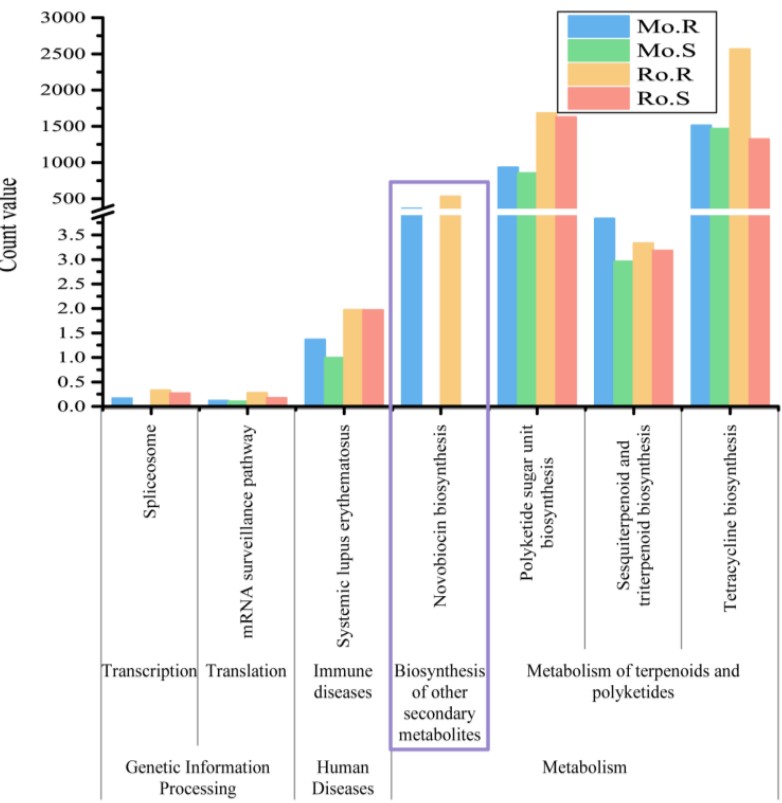

**Figure 10.** The column chart of KEGG enrichment terms of four groups. Purple frame box is the abundance of novobiocin biosynthesis in four groups.

## 4. Discussion

### 4.1. Effect of Severe RKN Disease on the Bacterial Diversity in Peanut Bulk Soil

The structure and function of the root microbiome play an important role in plant immunity and development and are closely related to plant health [40,41]. Some studies have shown that the suppression and outbreak of soilborne diseases are closely related to soil bacterial communities [42,43]. The increase of bacterial quantity and community structure diversity in soil is one of the main reasons to inhibit soilborne plant diseases [44]. Plants regulate their microbiome under biological stresses [45,46]. Our study found that the bacterial diversity of peanut bulk soil in the Ro site was higher than that in the Mo site; and RKN susceptible peanut showed a downward trend compared with RKN resistant peanut at the same planting site, but the difference was not significant (Figure 2). This result is consistent with previous studies that planting area can have a significant impact on bacterial community composition [47,48]. The results of this study supported that the bacterial diversity in bulk soil of RKN susceptible peanut was decreased compared with that of RKN resistant peanut.

We also found that *Acidobacteria*, *Proteobacteria*, *Actinobacteria*, *Planctomycetes*, *Chloroflexi*, *Firmicutes*, and *Bacteroidetes* were the most dominant phyla in peanut fields (Figure 3), which was broadly consistent with previous reports on peanut rhizosphere soil microbiota [22]. It was also found that samples at the same planting site were clustered together by β-diversity analysis (Figure 4). Distinct differences in bacterial communities can be observed between the two different sites, which indicates that cropping pattern or the severity of RKN disease has more influence on a bacterial community than peanut genotypes. It also been found that the abundance of *Actinobacteria*, *Planctomycetes*, and *Chloroflexi* in RKN-resistant peanut bulk soil, and similarly, Cao found that *Actinobacteria* and *Planctomycetes* were also present in healthy tobacco soils compared to RKN-susceptible tobacco soils [49].

### 4.2. Differential Bacterial Communities between Peanut Bulk Soils of Resistant and Susceptible to RKN

The result showed that *Sulfuricellaceae* at the family level was detectable as a specific biomarker in the bulk soil of RKN susceptible peanut compared to RKN resistant peanut at the same planting site (Figure 6). All members of *Sulfuricellaceae* utilize inorganic sulfur compounds as their energy source and use oxygen or nitrate as terminal electron acceptors for respiration. Its subordinate genus *Sulfuricella* [50,51] is also an indicator of a biomarker in Mo.S compared to Mo.R, and its subordinate genus *Sulfuriferula* is an indicator of a biomarker in Ro.S compared to Ro.R. It seems that the sulfur-like bacteria in the bulk soil of RKN-susceptible peanut are more prominent compared to that of RKN-resistant peanut.

*Singulisphaera* at the genus level was recognized as the specific biomarker in the bulk soil of RKN-resistant peanut compared to RKN-susceptible peanut at the same planting site (Figure 6). Representatives of this *Singulisphaera* genus are common inhabitants of soils and wetlands [52]. The *Singulisphaera* genus showed remarkable responses to pectin and xylan [53]. Moreover, xylan and pectin are important components of plant cell wall polysaccharides [54], which can prevent invasion and colonization of pathogenic microorganisms [55]. Although *Singulisphaera* seems to be inextricably related to peanut anti-KNF, further study is needed to provide more information on the mechanism.

### 4.3. The Role of Environment Factors in Bacterial Communities of Peanut Bulk Soil

Soil physicochemical properties are closely related to the bacterial communities of bulk soil [56]. Soil physical and chemical properties have a direct regulatory effect on plant root microenvironment and affect the composition and structure of root bacterial communities [57]. In this study, soil physical and chemical properties displayed an important factor affecting soil bacterial community structure (Figures 7 and 8), which was consistent with previous studies. In addition, we found that AP, AN, OM, and pH made a positive contribution to the OTUs of bacteria, while AK made a negative contribution; pH, OM, AN, and AP all had extremely significant differences along with bacteria *Acidibacter ferrireducens*, *Acidicapsa acidisoli*, and *Acidiferrimicrobium australe* (Figures 7 and 8).

### 4.4. Bacterial Potential Function in Peanut Bulk Soil

Soil microorganisms are an important part of the farmland ecosystem, which can promote the recycling of matter and energy in soil, especially the recycling and transformation of nutrient elements [58]. In our study, the highest functional enrichment of bacterial communities in peanut bulk soil was the first level of metabolism, and the second level xenobiotic biodegradation and metabolism. Seven metabolic pathways were selected as they were all increased in the resistant peanut bulk soil compared to the susceptible at the same planting site. Notably, we found that the novobiocin biosynthesis pathway differed 300-fold between resistant and susceptible peanut. It is presumed that RKN-resistant peanut could be involved in the biosynthesis of some novobiocin.

### 4.5. Beneficial and Harmful Bacteria in Peanut Bulk Soil

Beneficial microorganisms in soil can not only promote the transformation of soil organic matter into nutrients that can be absorbed and utilized by plants and improve the soil microecological environment but also produce a variety of bacteriostatic or bactericidal active substances, enhance the resistance of crops to a variety of diseases, and reduce the incidence of soilborne diseases [59]. Nitrifying has a profound effect on the form of mineral nitrogen that plants take up, use, and retain in the soil or lose to the environment [60]. Other beneficial bacteria had similar results, such as *Planctomyces*, *Gemmata*, *Flavisolibacter* [61]. It was found that *Arthrobacter* is a beneficial bacterium in corn fields [62], *Inquilinus* has the function of promoting the growth of ginseng [63], and *Nocardioides* belonging to the phylum actinomycetes can promote the growth of ginseng root [64]. Harmful bacteria, which have the opposite function of beneficial bacteria to plants, also deserve our attention. Literature stated that the relative abundance of beneficial bacteria such as *Sphingomonas*,

*Pseudomonas,* and *Aspergillus* increased significantly, while the relative abundance of the pathogen *Pythium* decreased significantly after crop rotation [65]. *Rhodoplanes* and *Kaistobacter* were the main bacteria in healthy roots of American ginseng, while *Sphingobium* was the main bacterial group in rotten roots [66]. In soil microorganisms, there are many promoting bacterial populations for plant growth and disease control, which are known as plant growth-promoting rhizobacteria (PGPR). PGPR has a certain biological control effect on soil-pathogenic microorganisms and nonparasitic rhizosphere harmful microorganisms [67]. The relative abundance of beneficial bacteria antagonistic to pathogenic bacteria, such as *Planctomyces*, *Bradyrhizobium*, and *Burkholderia* increased significantly, resulting in a low incidence under pineapple−banana rotation [68]. *Bacillus* bacterial agents can increase the abundance of the beneficial bacteria *Nitrospirae*, *Variovorax*, *Rhodanobacter*, *Nitrosospira*, *Rhodopseudomonas,* and *Mesorhizobium* [69]. It also showed that the family *Pseudomonadaceae* is beneficial to the control of root-knot nematodes [70].

Moreover, some studies have shown that *Bradyrhizobium*, *Rhizobia*, *Burkholderia*, and *Achromobacter* have the potential to biofix nitrogen with cowpea roots [71]. Non-pathogenic *Pseudomonas fluorescens* (WCS417r) and *Moringa oleifera* leaf extracts were effective against wheat aphids [70]. *Stenotrophomonas Maltophilia*, *Serratia Plymuthica*, *Pseudomonas Trivialis*, *P. Fluorescens*, *B. subtilis,* and *Burkholderia cepacia* can produce volatile organic compounds (VOCs) that inhibit the growth of plant pathogenic fungi hyphae [16]. In the research of biological control of plant diseases, the biological control of soilborne diseases has made great achievements. The bacteria used for biocontrol mainly belong to *Trichonderma*, *Streptomyces*, *Gliocladium*, *Bacillus*, *Pseudomonas*, *Agrobacterium*, *Flavobacter,* and *Enterobacter* [72,73].

Referring to the beneficial and harmful bacteria of other plants studied by previous reports, in this study, 11 candidate beneficial bacteria and 10 harmful bacteria were obtained for peanut resistance to RKN (Figure 9). *Burkholderia cepacia* accounted for the largest proportion of beneficial bacteria, and it may benefit from *Burkholderia cepacia* producing volatile organic compounds (VOCs) that inhibit the growth of plant pathogenic fungi hyphae [16]. *Burkholderia cepacia* could be used as a biological agent for peanut resistance to RKN in the future. *Bradyrhizobium* is generally beneficial for plants that can fix nitrogen, and it is also beneficial and antagonistic to pathogenic bacteria in banana rotation [68]. However, in our study, *Bradyrhizobium valentinum* is the largest percentage of potentially harmful bacteria for peanut resistance to RKN. It may be two-sided: it can fix nitrogen and is also pathogenic to peanut resistance to RKN.

## 5. Conclusions

Our findings strongly indicate that the planting site has more influence on the bacterial community of peanut bulk soil than the peanut genotype. *Singulisphaera* at the genus level was a biomarker in the bulk soil bacteria of RKN-resistant peanut compared to RKN-susceptible peanut in the same planting site and *Sulfuricellaceae* at the family level was detected to be a biomarker in that of RKN susceptible peanut. AP, AN, OM, and pH made a positive contribution to the OTUs of bacteria, while AK made a negative contribution. All pH, OM, AN, and AP had extremely significant differences on *Acidibacter ferrireducens*, *Acidicapsa acidisoli*, and *Acidiferrimicrobium australe.* The function of the novobiocin biosynthesis pathway plays an important role in peanut resistance to RKN. A total of 11 candidate-beneficial and 10 harmful bacteria were obtained for peanut resistance to RKN, and *Burkholderiacepacia*, as a beneficial bacterium against RKN in peanut could be used as a potential bioagent in the future.

These results highlight the significance of planting site, specific bacterial taxa, soil properties, and functional pathways in peanut resistance to RKN. The identification of candidate beneficial bacteria, including *Burkholderia epacia*, suggests the possibility of utilizing them as bioagents in future for RKN management strategies in peanut.

**Supplementary Materials:** The following supporting information can be downloaded at: https://www.mdpi.com/article/10.3390/agronomy13071803/s1, Figure S1: The Venn diagram of OTUs cluster in four peanut bulk soil groups. Figure S2: Cladogram showing specific phylotypes of peanut bulk soil between resistant or susceptible to RKN in Mo and Ro sites; Table S1: Preprocessing statistics and quality control of the Data. Table S2: ANOVA of Simpon and ACE index. Table S3: The table of Simpson and ACE indices in four groups. Table S4: Distribution and abundance of taxa at the phylum level. Table S5: Distribution and abundance of taxa at the class level; File S1: Details of OTUs. File S2: Details of the genus level of each group of bacterial taxa. File S3: Details of KEGG metabolic pathways the bacteria annotated to.

**Author Contributions:** Conceptualization, W.W. and M.Y.; methodology, L.W., Y.R., X.Z., F.Z., S.L. and M.Y.; software, L.W.; validation, Q.W. and M.Y.; formal analysis, L.W.; data curation, L.W.; writing—original draft preparation, L.W.; writing—review and editing, L.W., Y.R., G.C., Q.W., C.W., L.S., W.W. and M.Y.; visualization, L.W. and G.C.; supervision, M.Y. All authors have read and agreed to the published version of the manuscript.

**Funding:** This work was financially supported by the Natural Science Foundation of Shangdong Province, China (ZR2020QC121), the National Natural Science Foundation of China (31471533), the Key Research and Development Project of Shandong Province (2020LZGC001), Agricultural Scientific and Technological Innovation Project of Shandong Academy of Agricultural Sciences (CXGC2023A06, CXGC2021B33) and the Qingdao People's Livelihood Science and Technology Program, China (20-3-4-26-nsh).

**Data Availability Statement:** Most of the collected data are contained in the tables and figures in the manuscript and Supplementary Materials.

**Acknowledgments:** The authors thank Novogene Co. (Beijing, China) for data processing and Gene Denovo Co. (Guangzhou, China) for the assistance with data processing and bioinformatics analysis.

**Conflicts of Interest:** The authors declare that they have no known competing financial interests or personal relationships that could have appeared to influence the work reported in this paper.

## Abbreviations

OTU: Operational taxonomic unit; RKN: Root-knot nematode; Mo.R: Bulk soil of HY9810 (R-cultivar) which is resistant to RKN at Mo site; Mo.S: Bulk soil of HY20 (S-cultivar)which is susceptible to RKN at Mo site; Ro.R: Bulk soil of HY9810 (R-cultivar)which is resistant to RKN at Ro site; Ro.S: Bulk soil of HY20 (S-cultivar) which is susceptible to RKN at Ro site; LEfSe: LDA Effect Size; AN: Alkali-hydrolyzed nitrogen; AP: Available phosphorus; AK: Available potassium; OM: organic matter; ANOVA: Analysis of variance; PCA: Principal component analysis; PCoA: Principal coordinate analysis.

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
