# Peer review of "Effect of Root-Knot Nematode Disease on Bacterial Community Structure and Diversity in Peanut Fields"

_agronomy, doi:10.3390/agronomy13071803_

Round 1

Reviewer 1 Report

Abstract is far too long. Too much detail in terms of results. The abstract is supposed to briefly introduce the work and encourage the reader to read the whole manuscript. There is no need for a detailed description of the results. And the purpose of the research is not stated to be precisely about peanut fields. The abstract is almost as long as the whole introduction, which is incorrect.

Please correct the citation in the text - the paper numbers overlap the text.

L59-60: please provide references to support this statement - " The more diverse and stable the bacterial community is, the higher the 59 soil fertility is. It is better for the growth and development of the plants." - I am not aware of a paper that unequivocally posits such a conclusion.

L62-64: since "studies" please provide more than 1 reference

L65: scholars believe? What kind of wording is this? Scientists make theses, hypotheses, suggest, assume... but believe??

L69: Pseudomonas should be written in italics

L77-78: we utilized - in this context should be used....

In Sample collection it is very hard to read about the fields themselves. Maybe it is worth starting a map, a diagram of the experience? Also, a clear indication of the gap between objects should be provided.

L91: Add GPS coordinates

L96: but when were these samples taken? At what time of year? What were the weather conditions? How long was the peanut crop there, was it fertilised, irrigated? At what point in the growth of the plants were the samples taken? What length of roots do the nuts have, that only 0-20 cm were taken? The rhizosphere is the rooting soil, not "next to" the plant. Also, there is no information on what kind of soil it was. Type, classification?

L108: in spite of everything, the method of isolation should be given, because it influences the results

L115: why OTU and not ASV?

L118: references to R studio

L123: have the sequences been deposited in a public database e.g. NCBI? If yes, please provide the project code, if no.

L125-129: please reword this section and provide references for each analysis. What was the ratio of water to soil in the pH determination?

Figure 3: nine readable to me. 2 separate ones would be better.

L159: there is no Tukey test mentioned in the methodology. There is only LSD.

Figure 2, 3: what taxonomic level was used for analysis?

L192: UPGMA is not in the methodology either.

In L136 there is a subsection 3.1.1. The obtained total sequence data, and in L194 it appears again 3.1.1. Bacterial diversity. Something is not right here. Especially since Alpha diversity (L155-171) and Beta diversity (L175-188) are described earlier. Please reword and correct the order. Also, in my opinion L136-157 is redundant.

L169-171: redundant summary, here we describe the results, conclusions and discussion should go on

L171: than genotypes of what? It should be added that nuts.

L198-207: the names of phyla are not in italics

L251-252: what does 'other' mean in the diagrams?

L254: LEfSe is not in the methodology either; for people outside the topic LDA scores say absolutely nothing.

Figure 5: in my version, illegible captions on the graph.

Figure 6b: how was this calculated?

L298: in the sense that you added up or averaged the results for all the samples?

L313: please explain what *, **, *** are in the graph

L316: please refer to these literature data and write a little more about the criterion for selecting these and not other bacteria. And what do the authors mean by "beneficial" or "harmful"?

Figure 8: in my case e.g. illegible lettering

L335: this is also not in the methodology.

L354: this is a column chart rather than a histogram, it does not show the distribution of a single characteristic

Figure 9: what does "other secondary metabolites" mean? i.e. which ones?

L358-360: references?

L371-385: own results, not discussion

L530-543: to deleted

References: out of 56 papers, only 10 are from the last 5 years (2018-2023), while as many as 12 are below 2005 refs. 2, 3, 4 (1995!), 9, 15, 22, 23, 37,38,50,55 (1994!), 56. This needs to be corrected and more recent literature selected.

The language of the entire paper needs to be corrected. Sentences are badly compounded, too long sentences, lay words appear. Some words are used inappropriately e.g. L65, L77 etc.

Overall the paper is hard to read.

Author Response

Dear reviewer:

Thank you very much for your valuable comments on our manuscript, which will be of great benefit to improve our paper. The following are the answers to the relevant comments.

Open Review

Quality of English Language

( ) I am not qualified to assess the quality of English in this paper
( ) English very difficult to understand/incomprehensible
(x) Extensive editing of English language required
( ) Moderate editing of English language required
( ) Minor editing of English language required
( ) English language fine. No issues detected

Yes

Can be improved

Must be improved

Not applicable

Does the introduction provide sufficient background and include all relevant references?

( )

( )

(x)

( )

Are all the cited references relevant to the research?

( )

( )

(x)

( )

Is the research design appropriate?

( )

( )

(x)

( )

Are the methods adequately described?

( )

( )

(x)

( )

Are the results clearly presented?

( )

(x)

( )

( )

Are the conclusions supported by the results?

( )

(x)

( )

( )

Comments and Suggestions for Authors

Abstract is far too long. Too much detail in terms of results. The abstract is supposed to briefly introduce the work and encourage the reader to read the whole manuscript. There is no need for a detailed description of the results. And the purpose of the research is not stated to be precisely about peanut fields. The abstract is almost as long as the whole introduction, which is incorrect.

Answer: Abstract has been streamlined. And it has been modified in the uploaded version.

Please correct the citation in the text - the paper numbers overlap the text.

Answer: References in the paper have been processed in endnote. It has been modified in the uploaded version.

L59-60: please provide references to support this statement - " The more diverse and stable the bacterial community is, the higher the 59 soil fertility is. It is better for the growth and development of the plants." - I am not aware of a paper that unequivocally posits such a conclusion.

Answer: This sentence is not strict enough and has been modified in the uploaded version.

L62-64: since "studies" please provide more than 1 reference

Answer: The number of references has been increased and showed in the uploaded version.

L65: scholars believe? What kind of wording is this? Scientists make theses, hypotheses, suggest, assume... but believe??

Answer: Modified. It has been modified in the uploaded version.

L69: Pseudomonas should be written in italics

Answer: Modified. It has been modified in the uploaded version.

L77-78: we utilized - in this context should be used....

Answer: It has been modified in the uploaded version.

In Sample collection it is very hard to read about the fields themselves. Maybe it is worth starting a map, a diagram of the experience? Also, a clear indication of the gap between objects should be provided.

L91: Add GPS coordinates

Answer: GPS information has been added to the 2.1. Sample collection and processing.

L96: but when were these samples taken? At what time of year? What were the weather conditions? How long was the peanut crop there, was it fertilised, irrigated? At what point in the growth of the plants were the samples taken? What length of roots do the nuts have, that only 0-20 cm were taken? The rhizosphere is the rooting soil, not "next to" the plant. Also, there is no information on what kind of soil it was. Type, classification?

Answer: The information about peanuts at the time of sampling has been added in 2.1. Sample collection and processing.

L108: in spite of everything, the method of isolation should be given, because it influences the results

Answer: It has been modified in the uploaded version.

L115: why OTU and not ASV?

Answer: ASV has many points compared with OTU, but ASV may reject the presence of some very low abundance species in the sample, and in addition ASV needs to build a suitable error model to accurately detect the wrong sequences. Taking into consideration, we finally selected OTU.

L118: references to R studio

Answer: References have been added in the uploaded version.

L123: have the sequences been deposited in a public database e.g. NCBI? If yes, please provide the project code, if no.

Answer: We are now uploading the data information to NCBI, there is no the project code yet.

L125-129: please reword this section and provide references for each analysis. What was the ratio of water to soil in the pH determination?

Answer: This section has been reworked and provided as a reference for each analysis. The ratio of water to soil in the pH determination were 1: 2.5. It has been modified in the uploaded version.

Figure 3: nine readable to me. 2 separate ones would be better.

Answer: It was an author error that made this section difficult to read and has been revised. Due to the large number of charts, the two figures were combined in consideration of the layout.

L159: there is no Tukey test mentioned in the methodology. There is only LSD.

Answer: Modified at 2.4. Statistical analysis.

Figure 2, 3: what taxonomic level was used for analysis?

Answer: Figure 2 is based on OTU level. Figure 3a at the OTU level and 3b at the family level.

L192: UPGMA is not in the methodology either.

Answer: Modified at methodology.

In L136 there is a subsection 3.1.1. The obtained total sequence data, and in L194 it appears again 3.1.1. Bacterial diversity. Something is not right here. Especially since Alpha diversity (L155-171) and Beta diversity (L175-188) are described earlier. Please reword and correct the order. Also, in my opinion L136-157 is redundant.

Answer: The preceding and following data have been verified and corrected. L136-157 is a general summary of the obtained data, and it is still necessary to retain.

L169-171: redundant summary, here we describe the results, conclusions and discussion should go on

Answer: It has been modified.

L171: than genotypes of what? It should be added that nuts.

Answer: The sentence has been deleted.

L198-207: the names of phyla are not in italics

Answer: Italicized.

L251-252: what does 'other' mean in the diagrams?

Answer: It has been modified in the uploaded version.

L254: LEfSe is not in the methodology either; for people outside the topic LDA scores say absolutely nothing.

Answer: Added in the methods section.

Figure 5: in my version, illegible captions on the graph.

Answer: PDF version is more clear have been uploaded.  

Figure 6b: how was this calculated?

Answer: It has been modified in the uploaded version.

L298: in the sense that you added up or averaged the results for all the samples?

Answer: In this sense, we averaged the results for all the samples.

L313: please explain what *, **, *** are in the graph

Answer: It has been modified in the uploaded version.

L316: please refer to these literature data and write a little more about the criterion for selecting these and not other bacteria. And what do the authors mean by "beneficial" or "harmful"?

Answer: Definitions have been made in the paper, and references have been added.

Figure 8: in my case e.g. illegible lettering

Answer: Word and PDF versions have been uploaded.  

L335: this is also not in the methodology.

Answer: This has been added to the method.

L354: this is a column chart rather than a histogram, it does not show the distribution of a single characteristic

Answer: It has been changed to column chart.

Figure 9: what does "other secondary metabolites" mean? i.e. which ones?

Answer: In the present study, other secondary metabolites mainly included Betalain biosynthesis, Clavulanic acid biosynthesis, Flavonoid biosynthesis, Isoflavonoid biosynthesis, Neomycin, kanamycin and gentamicin biosynthesis, Novobiocin biosynthesis, Penicillin and cephalosporin biosynthesis, Streptomycin biosynthesis, Tropane, piperidine and pyridine alkaloid biosynthesis, and so on.

L358-360: references?

Answer: References have been added.

L371-385: own results, not discussion

Answer: Some results were deleted and added some discussions.

L530-543: to deleted

Answer: Deleted.

References: out of 56 papers, only 10 are from the last 5 years (2018-2023), while as many as 12 are below 2005 refs. 2, 3, 4 (1995!), 9, 15, 22, 23, 37,38,50,55 (1994!), 56. This needs to be corrected and more recent literature selected.

 Answer: Seventeen new references were added due to the necessity of revising the article, and most of them are after 2018.

Comments on the Quality of English Language

The language of the entire paper needs to be corrected. Sentences are badly compounded, too long sentences, lay words appear. Some words are used inappropriately e.g. L65, L77 etc.

Answer: A lot of modifications have been made.

Overall the paper is hard to read.

 Answer: We have asked my English colleagues to help with the revision.

Thank you and best regards.

Yours sincerely,

Reviewer 2 Report

This study used high-throughput sequencing technology to try to reveal the relationship between microbes and root-knot nematode disease in peanut rhizosphere soil, and obtained some results that will help to use beneficial microbes to improve peanut resistance to root-knot nematodes. This work is interesting, but there are some issues that need to be clearly explained before the article is accepted.

Firstly, for the peanut cultivars HY9810 and HY20, whether there is any relevant literature or data to prove their level of resistance to root-knot nematodes, and the species of root-knot nematode also needs to be clarified. Different species may be present in different sites and this has a direct impact on the severity of the disease.

Secondly, with regard to the trial site, the Mo site was a peanut-maize-wheat rotation with a low intensity of root-knot nematodes, whereas the Ro site was a peanut continuous crop with a high intensity of root-knot nematodes. The authors compared microbial differences in the rhizosphere soils of the same cultivar at two different sites, how could distinguish whether the differences were due to the cropping pattern or influenced by the root-knot nematodes in the soil?

The time of soil sampling also needs to be specified- was it at the seedling, growing or harvest stage of the peanut? What was the extent of root-knot nematode infestation at the time of sampling? Microbial structure and diversity can change at different times.

Minor editing of English language required

Author Response

Dear reviewer:

Thank you very much for your suggestions on the manuscript. Here are the answers to the questions.

Open Review

Quality of English Language

( ) I am not qualified to assess the quality of English in this paper
( ) English very difficult to understand/incomprehensible
( ) Extensive editing of English language required
( ) Moderate editing of English language required
(x) Minor editing of English language required
( ) English language fine. No issues detected

Comments and Suggestions for Authors

This study used high-throughput sequencing technology to try to reveal the relationship between microbes and root-knot nematode disease in peanut rhizosphere soil, and obtained some results that will help to use beneficial microbes to improve peanut resistance to root-knot nematodes. This work is interesting, but there are some issues that need to be clearly explained before the article is accepted.

Firstly, for the peanut cultivars HY9810 and HY20, whether there is any relevant literature or data to prove their level of resistance to root-knot nematodes, and the species of root-knot nematode also needs to be clarified. Different species may be present in different sites and this has a direct impact on the severity of the disease.

Answer: Pictures of two peanut varieties infected with root-knot nematodes are being collected and added.

Secondly, with regard to the trial site, the Mo site was a peanut-maize-wheat rotation with a low intensity of root-knot nematodes, whereas the Ro site was a peanut continuous crop with a high intensity of root-knot nematodes. The authors compared microbial differences in the rhizosphere soils of the same cultivar at two different sites, how could distinguish whether the differences were due to the cropping pattern or influenced by the root-knot nematodes in the soil?

Answer: The same degree of root knot nematode of the same peanut variety, indicating that the differences were due to the cropping pattern; we also compared microbial differences in the rhizosphere soils of the different cultivars at the same sites, this result then completely explained the differences were caused by root-knot nematodes.

The time of soil sampling also needs to be specified- was it at the seedling, growing or harvest stage of the peanut? What was the extent of root-knot nematode infestation at the time of sampling? Microbial structure and diversity can change at different times.

Answer: The time of soil sampling was done at the time of mature and full-fruited and the root-knot nematode has fully developed by this time. It has been modified in the uploaded version.

Thank you and best regards.

Yours sincerely,

Reviewer 3 Report

The research entitled (Effect of Root-Knot Nematode Disease on Microbial Community Structure and Diversity in Peanut Rhizosphere) aimed to systematically study the microbial diversity change of peanut rhizosphere soil after infecting the RKN.

1.     Introduction part is very poor and needs more information and recent references.

2.     What is the two planting sites' environmental condition during the disease period?

3.     Why did the authors study only the soil bacteria and discard the fungal biota?

4.     What is the soil moisture content of the samples taken?

5.     Its more preferable to add infection and nematode photos.

6.     Why did the relative abundance of Actinobacteria, Planctomycetes and Chloroflexi increase? Is there any relation between them and the nematode presence?

7.     The reference part needs revision one by one.

The manuscript needs English editing and grammar corrections

Author Response

Dear reviewer:

Thank you very much for your suggestions and recommendations on the manuscript. Here are the answers to the relevant questions.

Open Review

Quality of English Language

( ) I am not qualified to assess the quality of English in this paper
( ) English very difficult to understand/incomprehensible
( ) Extensive editing of English language required
(x) Moderate editing of English language required
( ) Minor editing of English language required
( ) English language fine. No issues detected

Yes

Can be improved

Must be improved

Not applicable

Does the introduction provide sufficient background and include all relevant references?

( )

( )

(x)

( )

Are all the cited references relevant to the research?

( )

(x)

( )

( )

Is the research design appropriate?

( )

(x)

( )

( )

Are the methods adequately described?

( )

(x)

( )

( )

Are the results clearly presented?

( )

(x)

( )

( )

Are the conclusions supported by the results?

( )

(x)

( )

( )

Comments and Suggestions for Authors

The research entitled (Effect of Root-Knot Nematode Disease on Microbial Community Structure and Diversity in Peanut Rhizosphere) aimed to systematically study the microbial diversity change of peanut rhizosphere soil after infecting the RKN.

  1. Introduction part is very poor and needs more information and recent references.

Answer: The Introduction section has been revised and is available for review.

  1. What is the two planting sites' environmental condition during the disease period?

Answer: The section 2.1. Sample collection and processing has been modified in detail to add detailed information on sampling.

  1. Why did the authors study only the soil bacteria and discard the fungal biota?

Answer: This manuscript focuses on the bacterial aspects, and the fungal studies will be used as a later research direction.

  1. What is the soil moisture content of the samples taken?

Answer: Samples were taken in mid-September, and the soil moisture content was visually estimated to be about 10 %.

  1. Its more preferable to add infection and nematode photos.

Answer: Photos of two varieties infected with root-knot nematodes have been added.

  1. Why did the relative abundance of Actinobacteria, Planctomycetes and Chloroflexi increase? Is there any relation between them and the nematode presence?

Answer: Actinobacteria and Planctomycetes were also present in healthy plant soils compared to RKN-susceptible plant (Cao, 2015). The relationship between RKN and Actinobacteria and Planctomycetes needs to be proved by our further study.

  1. The reference part needs revision one by one.

Answer: References have been checked individually.

Thank you and best regards.

Yours sincerely,

Round 2

Reviewer 2 Report

The title of the article needs to be changed as only the bacterial microbiota was studied.

Average

Author Response

Dear reviewer:

Thank you very much for your comments and suggestions on the manuscript again. Here are the answers to the question.

Open Review

Quality of English Language

( ) I am not qualified to assess the quality of English in this paper
( ) English very difficult to understand/incomprehensible
( ) Extensive editing of English language required
( ) Moderate editing of English language required
(x) Minor editing of English language required
( ) English language fine. No issues detected

Comments and Suggestions for Authors

The title of the article needs to be changed as only the bacterial microbiota was studied.

Answer: The title was changed to “Effect of Root-Knot Nematode Disease on Bacterial Community Structure and Diversity in Peanut Fields” after discussion.  

Moreover, we have revised and verified the data and language in other parts of the manuscript.

Thank you and best regards.

Yours sincerely,

Reviewer 3 Report

1.     Introduction part is still very poor the authors add references without information’s.

2.     What is the two planting sites environmental condition during the disease period?

The author's response was section 2.1. Sample collection and processing has been modified in detail to add detailed information on sampling. However, I found no environmental conditions (temperature, humidity, wind speed, etc) in section 2.1.?

Revision by a native speaker is recommended

Author Response

Dear reviewer:

Thank you very much for your comments and suggestions on the manuscript again. Here are the answers to the question.

Open Review

Quality of English Language

( ) I am not qualified to assess the quality of English in this paper
( ) English very difficult to understand/incomprehensible
( ) Extensive editing of English language required
(x) Moderate editing of English language required
( ) Minor editing of English language required
( ) English language fine. No issues detected

Comments and Suggestions for Authors

  1. Introduction part is still very poor the authors add references without information’s.

Answer: The introduction has been reworked. All the references have been reordered and checked. Moreover, we have revised and verified the data and language in other parts of the manuscript.

  1. What is the two planting sites environmental condition during the disease period?

The author's response was section 2.1. Sample collection and processing has been modified in detail to add detailed information on sampling. However, I found no environmental conditions (temperature, humidity, wind speed, etc) in section 2.1.?

Answer: We are so sorry about this. We cannot provide environmental information, because we just recorded the sampling information at that time. Sorry. We will record relevant environmental information next time.

Thank you and best regards.

Yours sincerely,